# Live-Cell Imaging of Physiologically Relevant Metal Ions Using Genetically Encoded FRET-Based Probes

**DOI:** 10.3390/cells8050492

**Published:** 2019-05-22

**Authors:** Helmut Bischof, Sandra Burgstaller, Markus Waldeck-Weiermair, Thomas Rauter, Maximilian Schinagl, Jeta Ramadani-Muja, Wolfgang F. Graier, Roland Malli

**Affiliations:** 1Gottfried Schatz Research Center, Chair of Molecular Biology and Biochemistry, Medical University of Graz, Neue Stiftingtalstraße 6/6, 8010 Graz, Austria; sandra.burgstaller@medunigraz.at (S.B.); markus.weiermair@medunigraz.at (M.W.-W.); thomas.rauter@medunigraz.at (T.R.); maximilian.schinagl@medunigraz.at (M.S.); jeta.ramadani@medunigraz.at (J.R.-M.); wolfgang.graier@medunigraz.at (W.F.G.); roland.malli@medunigraz.at (R.M.); 2BioTechMed Graz, Mozartgasse 12/II, 8010 Graz, Austria

**Keywords:** FRET, fluorescent protein, FP, fluorescence microscopy, genetically encoded probes, imaging, Ca^2+^, Cu^+^, Cu^2+^, Fe^2+^, Fe^3+^, K^+^, Mg^2+^, Mn^2+^, Zn^2+^

## Abstract

Essential biochemical reactions and processes within living organisms are coupled to subcellular fluctuations of metal ions. Disturbances in cellular metal ion homeostasis are frequently associated with pathological alterations, including neurotoxicity causing neurodegeneration, as well as metabolic disorders or cancer. Considering these important aspects of the cellular metal ion homeostasis in health and disease, measurements of subcellular ion signals are of broad scientific interest. The investigation of the cellular ion homeostasis using classical biochemical methods is quite difficult, often even not feasible or requires large cell numbers. Here, we report of genetically encoded fluorescent probes that enable the visualization of metal ion dynamics within individual living cells and their organelles with high temporal and spatial resolution. Generally, these probes consist of specific ion binding domains fused to fluorescent protein(s), altering their fluorescent properties upon ion binding. This review focuses on the functionality and potential of these genetically encoded fluorescent tools which enable monitoring (sub)cellular concentrations of alkali metals such as K^+^, alkaline earth metals including Mg^2+^ and Ca^2+^, and transition metals including Cu^+^/Cu^2+^ and Zn^2+^. Moreover, we discuss possible approaches for the development and application of novel metal ion biosensors for Fe^2+^/Fe^3+^, Mn^2+^ and Na^+^.

## 1. Introduction

Vital cells tightly control their (sub)cellular ion distributions [1]. Alterations of the intracellular ion homeostasis are associated with severe dysfunctions and pathologies. Frequently, neurodegenerative processes, cancerous alterations and systemic diseases are linked to changes in the intracellular ion composition [2,3,4,5,6,7]. Especially the distribution of metal ions within cells is of great importance, as they can act as second messengers, modulate signaling pathways or directly affect enzyme functions due to their (in)direct involvement in enzyme-substrate interactions [2,8,9,10,11,12]. However, our current knowledge about intracellular ion concentrations in different organelles is quite limited mainly due to the lack of suitable quantification methods.

Besides the importance of an intact intracellular ion homeostasis, the extracellular ion composition is also of fundamental importance to maintain cellular physiology (Figure 1) [13]. Cells communicate with each other via their (local) environment, where minor changes of the ion composition within the extracellular fluid (ECF) can affect cell activities and even promote the development of severe diseases [4,14,15].

Thereby, especially in the surrounding of excitable cells or cells undergoing cell death, the ion composition of the ECF is drastically affected [4,16,17]. Notably, ion changes in the ECF impact the intracellular ion homeostasis, which might lead to compensatory mechanisms of cells to maintain their intracellular ion composition [18,19]. As an example, traumatic brain injuries are associated with extracellular potassium ion concentration ([K^+^]) alterations in the brain. These extracellular potassium ion (K^+^) fluctuations have been linked with a poor survival of these patients, due to an increase of the intracranial pressure, caused by either cell swelling or altered vasoreactivity in cerebral blood vessels [20]. Moreover, an intake of angiotensin-converting-enzyme (ACE) inhibitors by patients suffering from renal insufficiency was linked to the development of severe hyperkalemia, which in turn promotes the development of cardiac arrhythmias due to cardiac-myocyte depolarization [21,22,23]. 

Although measurements of the extracellular ion concentrations are feasible using clinic-chemical detection methods, their intra- and subcellular distributions and dynamics upon cell stimulation are difficult to capture [24]. The development of reliable detection methods for intracellular metal ions is inevitable to gain our understanding of the mechanisms ultimately causing diseases and may serve as an essential basis for the development and improvement of medical therapies [25].

Valuable tools for the performance of such measurements represent genetically encoded, fluorescent biosensors. The discovery and isolation of the green fluorescent protein (GFP) in 1962 by Shimomura et al. paved the way for the development of GFP-based probes and tools, which have boosted our understanding of cell biology as well as molecular and cellular (patho)physiology [26,27]. Nowadays, numerous differently colored fluorescent protein (FP) variants have been engineered [28]. The first successful design of a so-called genetically encoded probe (GEP) was achieved in 1997 by Miyawaki et al., who exploited Förster resonance energy transfer (FRET) occurring between a yellow and a cyan fluorescent protein (YFP, CFP) variant, fused to specific calcium ion (Ca^2+^) binding domains [27]. These probes were based on a conformational rearrangement upon Ca^2+^ binding, yielding an increase in FRET, while the donor fluorescence, i.e., CFP, decreases [27]. The development of this sophisticated probes, referred to as “*Cameleons*”, opened the door for many further FRET-based indicators that allow real-time recordings of cell signaling events [29,30,31,32]. Nowadays, a huge variety of GEPs is available, especially for metal ions including protein-based fluorescent sensors for Ca^2+^, copper ions (Cu^+^/Cu^2+^), magnesium ions (Mg^2+^), K^+^ and zinc ions (Zn^2+^) [30,33,34,35,36].

In principle, these probes mostly consist of CFP-YFP (Figure 2a) or GFP- red fluorescent protein (RFP) (Figure 2b), fused to specific ion binding domains [27,30,32]. Ion binding to the ion sensitive domain within the probe changes the distance of the two FPs located at the termini of the binding domain, which leads to alterations of the fluorescent properties (Figure 2) [37]. This design allows dynamic and ratiometric read-outs of cellular ion fluctuations, as upon excitation of the donor FP, two emission wavelengths are recorded simultaneously, showing opposite fluorescence alterations depending on the ion concentration (Figure 2c,d). The FRET/donor fluorescence ratio is directly proportional to the concentration of the ion (Figure 2c,d) and allows the investigation of intracellular ion dynamics with high spatial and temporal resolution upon cell treatment (Figure 3) [37]. 

Besides these FRET-based sensors, a broad range of different probes consisting of only one FP have been developed. These indicators feature some advantages over their FRET-based ancestors, including narrower spectral properties, small size and a huge dynamic range [38,39,40,41]. A detailed review focusing on diverse classes of genetically encoded probes is available [42]. Nonetheless, until today FRET-based probes represent the gold standard of GEPs, due to their ratiometric, and thus, easily quantifiable read-out [43]. However, single FP-based probes represent valuable tools to perform multicolor and –parameter imaging on the single cell level, often in combination with FRET-based probes [30,44]. Nevertheless, such measurements require a sophisticated experimental setup to properly distinguish and separate the spectral features of different probes. Several manuscripts focusing on the usage of genetically encoded probes from a rather technical point of view are available [42,45,46]. 

Notably, a simple fusion with an approved localization domain results in a, mostly, distinct subcellular distribution of these GEPs [47]. In contrast to most chemical ion indicators, the targeting of biosensors allows the visualization and quantification of ion signals on the subcellular level [30,32,48]. 

Here we review a selection of frequently used FRET-based GEPs, suitable for the high-resolution investigation of physiologically important, (sub)cellular alkali metals such as K^+^, alkaline earth metals including Mg^2+^ and Ca^2+^, and transition metal ions such as Cu^+^/Cu^2+^ and Zn^2+^. Additionally, we will focus on novel strategies for the design of probes, which might be suitable for the visualization of sodium ions (Na^+^), iron ions (Fe^2+^/Fe^3+^) and manganese ions (Mn^2+^) on the single cell level. 

## 2. Genetically Encoded Fluorescent Probes for Imaging the Alkali Metal Ion K^+^, Highly Desired and Freshly Introduced

Living cells maintain a steep K^+^ gradient across their plasma membrane, while K^+^ represents the most abundant intracellular cation [13]. Besides its fundamental involvement in the generation of the electrochemical gradient important for many cell types in order to remain excitable, K^+^ is essentially involved in the regulation of the cell volume, endo- and exocytosis or the prevention of apoptosis [49,50,51,52,53]. In addition, K^+^ is a central cofactor of several enzymes. The pyruvate kinase, the pyruvate dehydrogenase kinase, the diol- and the glycerol dehydratase or the ribokinase are only some of many examples, pointing to K^+^ as a master regulator of metabolic pathways [54,55,56,57,58,59]. Despite the important role of K^+^ in cell physiology, the development of K^+^ sensitive probes has also been hampered by the lack of specific K^+^ binding domains, suitable for the use within GEPs. Thus, several chemical fluorescent K^+^ indicators, based on crown ethers, applicable for the visualization of intracellular [K^+^] have been developed [60,61,62]. However, these K^+^ binding probes such as PBFI and Asante Potassium Green-1 (APG-1) suffer from poor specificity for K^+^ over Na^+^, require phototoxic excitation with UV-light or absolute quantifications of [K^+^] are difficult [60,61]. 

Recently, Ashraf and coworkers unraveled the function of a K^+^ binding protein, shortly Kbp (K^+^ binding protein), formerly YgaU, a protein of unknown function until then, found in *Escherichia coli (E. coli)* [63]. The expression of Kbp, consisting of a bacterial OsmY and nodulation (BON) domain and a lysine motife (LysM), was known to be promoted by the bacterial RNA polymerase sigma S (RpoS) regulon [64]. In principle, genes underlying the RpoS regulon in *E. coli* are associated with the stress response, thereby acting pro- and retroactive [65]. Within their study, Ashraf et al. demonstrated that Kbp represents a K^+^ binding protein in these bacteria, important to enable bacterial growth at extracellular [K^+^] above 1 mM [63]. Upon K^+^ binding, Kbp undergoes a huge conformational rearrangement, making it ideal for the design of GEPs for K^+^ sensing and imaging [30,63]. 

Based on Kbp, we and others have recently developed K^+^ sensitive FRET-based GEPs referred to as Genetically Encoded Potassium Ion Indicators (GEPIIs) [30,66]. The GEPIIs proved suitable for the determination of extra- and intracellular [K^+^]. Thereby, the GEPII variant containing the wild type Kbp, referred to as GEPII 1.0, showed a dissociation rate constant (K_D_) of ~ 0.8 mM when expressed in cells, which is suitable for the quantification of extracellular [K^+^] [30,67]. However, the high affinity of GEPII 1.0 is far from physiologic intracellular [K^+^] of ~ 140 mM [68]. Thus, for being able to measure intracellular K^+^ fluctuations, different approaches to desensitize the probe for K^+^ were tested, including the introduction of linkers of variable length between the BON and the LysM domain or the insertion of point mutations within the domains, respectively. The introduction of point mutations resulted in probes referred to as low charge (lc-) BON GEPII 1.0 and lc-LysM GEPII 1.0, which either totally lost their K^+^ sensitivity or showed much lower affinity for K^+^ than GEPII 1.0 [30].

Expression of lc-LysM GEPII 1.0 within various organelles of several cell lines allowed the quantification of subcellular [K^+^] within the cytosol, mitochondria and nucleus of these cells. Interestingly, cell line specific differences of the nuclear/cytosolic as well as mitochondrial/cytosolic [K^+^] have been described. These differences might represent a hallmark of cancerous alterations [30]. However, direct evidences for correlations of cancerous malignancies and the distribution of intracellular [K^+^] are still missing. Nonetheless, GEPIIs will significantly contribute to our understanding of the K^+^ distribution, its dynamics and the role of K^+^ in health and disease. In particular, K^+^ has recently been demonstrated to be involved in the onset and progression of tumor growth [4]. Direct evidences for an increased intracellular [K^+^] ([K^+^]_i_) are still missing, probably due to the lack of applicable detection methods for K^+^. GEPIIs could therefore represent valuable tools to correlate the [K^+^]_i_ with the degree of suppression of T-cell receptor (TCR) activation. 

Besides the quantification of [K^+^]_i_, such GEPIIs represent applicable tools for the visualization of K^+^ fluctuations over time. Recently, these FRET-based GEPs were simultaneously expressed in single cells with genetically encoded, single FP-based, red fluorescent Ca^2+^ indicators [41]. Cells expressing the probes were analyzed in response to cell depolarization (Figure 4a). Such experiments allowed the correlation of [K^+^]_i_ with intracellular Ca^2+^ alterations upon cell depolarization (Figure 4b) [30]. 

These genetically encoded K^+^ indicators have great potential to deepen our knowledge of the cellular K^+^ homeostasis on the level of different organelles. Furthermore, understanding [K^+^] distributions under conditions of health and disease may allow the development of novel therapeutic approaches. A selection of fluorescent K^+^ indicators suitable for intracellular K^+^ imaging is given in Table 1.

## 3. Genetically Encoded Fluorescent Probes for Alkaline Earth Metal Ions

### 3.1. Genetically Encoded Mg^2+^ Indicators, Sophisticated Tools, Rarely Applied 

Besides K^+^ as the most abundant intracellular monovalent cation, Mg^2+^ represents the most abundant intracellular divalent cation [69]. Mg^2+^ is fundamentally involved in a huge variety of essential cellular processes including transcription, cell replication, energy metabolism or the regulation of specific ion channels [70,71,72,73]. Besides, Mg^2+^ is a central cofactor of many enzymes and regulates especially the activity of glycolytic enzymes in the chelated form of MgATP^2−^ [74,75,76]. In addition, Mg^2+^ is considered to be involved in the intracellular signal transduction as a second messenger [77]. The total intracellular Mg^2+^ concentration ([Mg^2+^]_i_) ranges from 17–20 mM, while the free Mg^2+^ concentration was estimated to be around 1 mM [69]. Various hereditary disorders such as diminished kidney function, renal failure, muscle spasms and seizures have been linked to disorders in the Mg^2+^ homeostasis [78,79,80,81]. Additionally, the onset of cellular senescence and thus ageing-related diseases such as diabetes or cardiovascular disease have been linked to Mg^2+^ homeostasis alterations [82,83,84]. 

Considering the importance of intracellular Mg^2+^, scientists early aimed for the development of Mg^2+^ sensitive fluorescent indicators such as Mag-fura-2 [85]. However, the first FRET-based GEP sensitive for Mg^2+^ was introduced by Lindenburg et al. not earlier than 2013 [34]. They developed a Mg^2+^ sensitive GEP, based on the human centrin isoform HsCen3 [34]. In principle, centrins represent proteins associated with the centrosome of cells, important for centrosome duplication and separation [86]. It has been demonstrated earlier that HsCen3 has a mixed Ca^2+^ and Mg^2+^ binding site possessing high affinity at its N-terminus, while a second Ca^2+^ specific binding domain located at the C-terminus shows low Ca^2+^ specific affinity [87].

In the study of Lindenburg et al., truncation of the protein and the usage of the N-terminal fragment within an FP FRET-pair led to the construction of a probe sensitive for Mg^2+^, referred to as MagFRET. In the apo state, HsCen3 forms a molten globule, which folds into a compact EF-hand like structure upon Mg^2+^ fixation. The introduction of single point mutations within the EF-hand like motifs led to the construction of a huge variety of Mg^2+^ sensitive indicators, showing different Mg^2+^ affinities and different specificities over Ca^2+^ binding [34]. However, these probes have only rarely been used for the investigation of intracellular Mg^2+^ fluctuations on the level of individual cells.

Another approach of the development of genetically encoded Mg^2+^ indicators represents the magnesium ratiometric indicator for optical imaging (MARIO) [88]. The function of the FRET-based probe MARIO is based on the Mg^2+^ transporter CorA found in *E. coli*, undergoing a conformational rearrangement upon Mg^2+^ binding [89]. In bacteria, CorA mediates Mg^2+^ influxes and effluxes into and from the cells [90]. CorA localizes within the membrane of *E. coli* [90]. Nonetheless, structural data of CorA seem controversial, as older studies suggest that the majority of the protein is localized in the periplasm, with a small C-terminal part in the cytosol, while recent studies imply only a cytoplasmic localization [89,90]. For the generation of MARIO the original protein of CorA was modified by deleting the N-terminal region of the protein, as structural data suggested that this region is not involved in Mg^2+^ binding [88,89]. Additionally, polar-charged amino acids were replaced and random mutations were introduced. Performing these modifications of CorA and usage of the protein within a FP FRET-pair yielded a probe showing a lower K_D_ for Ca^2+^ (6.2 mM) than for Mg^2+^ (7.2 mM). Nonetheless, the Ca^2+^ sensitivity of this sensor is far from the physiological cytoplasmic range [91]. Using MARIO, the authors reported an increase in the free cytosolic Mg^2+^ concentration ([Mg^2+^]), which contributes to mitotic chromosome condensation [88]. However, further applications of MagFRET or MARIO have not been performed until today, thus, insights into (patho-)physiological alterations of the intra- and especially subcellular Mg^2+^ homeostasis remain mainly elusive. A selection of FRET-based genetically encoded Mg^2+^ indicators suitable for intracellular Mg^2+^ imaging is given in Table 2.

### 3.2. Genetically Encoded Ca^2+^ Indicators, A Huge Variety for An Ion with Versatile Roles

Ca^2+^ represents a central second messenger in living cells and is, thus, involved in countless signaling pathways [92]. Besides its involvement in the regulation of enzymatic activities as well as ion channels and ion pumps, Ca^2+^ fundamentally contributes to the regulation of gene transcription [93,94,95,96,97]. Additionally, Ca^2+^ participates in the contraction of all muscle cells, neurotransmitter and hormone release or fertilization [98,99,100,101]. Especially the mitochondrial and ER Ca^2+^ homeostasis and the Ca^2+^ concentration ([Ca^2+^]) within the organelles are tightly regulated [102,103,104]. The ER represents an intracellular Ca^2+^ storage with [Ca^2+^] up to 1 mM, which is important for the ER in order to retain its structure and allow protein folding by Ca^2+^ dependent chaperones such as BiP or the protein disulfide-isomerase (PDI) [105,106,107]. Ca^2+^ transfer from the ER towards the mitochondrial matrix is necessary to ensure physiological cell bioenergetics [104,108]. The mitochondrial matrix features a strongly negative membrane potential, thus, Ca^2+^ would rapidly accumulate within the matrix without the strict Ca^2+^ handling of the organelle, further leading to the induction of apoptosis, and finally cell death [109,110,111]. Ca^2+^ fluxes into the mitochondrial matrix are, thus, highly regulated and achieved via a whole protein ensemble, especially including the mitochondrial calcium uniporter (MCU), the mitochondrial calcium uptake 1 and 2 (MICU1/2) and the essential MCU regulator (EMRE). Detailed reviews about the regulation of mitochondrial Ca^2+^ uptake and homeostasis are available [102,112,113].

Considering the importance of Ca^2+^ as a ubiquitous intracellular messenger, the variety of Ca^2+^ sensitive sensors represents likely the most extensive one of all available indicators. Until today, one of the most prominent designs of a Ca^2+^ indicator suitable for the visualization of global intracellular Ca^2+^ alterations represents Fura-2, a ratiometric chemical fluorescent Ca^2+^ sensitive dye [114]. However, as all chemical fluorescent indicators, Fura-2 features some severe disadvantages in comparison to GEPs, including phototoxicity due to excitations with UV-light, difficult subcellular targeting and poor applicability in living animals. Thus, scientists aimed for the development of genetically encoded FRET-based GEPs. The first successful design of a FRET-based genetically encoded Ca^2+^ indicator succeeded in 1997 by Miyawaki et al., who introduced the first genetically encoded sensor of all time and paved the way for countless further developments [27]. By the fusion of a CFP and a YFP variant to calmodulin (CaM), a Ca^2+^ binding protein, and the myosin light chain kinase M13, a protein interacting with Ca^2+^ bound CaM, they yielded functional Ca^2+^ sensors referred to as Cameleons [27]. Nowadays, a huge variety of these Cameleons is available. Cameleons with different Ca^2+^ sensitivities for different cellular organelles with organelle specific [Ca^2+^] have been successfully generated [27,32,48,115,116,117,118,119,120,121]. The most frequently used probes thereby contain Ca^2+^ sensitive domains referred to as D1, D2, D3 or D4, which represent mutated variants of CaM and M13, respectively [27,32,48,115,116,117,118,119,120,121]. Using these domains, the visualization of intracellular relevant [Ca^2+^] became possible within the mitochondria, the nucleus, the cytosol, the ER, the Golgi apparatus or endosomes of single living cells, all possessing different [Ca^2+^] [27,32,33,48,115,116,117,118,119,120,121]. Besides FRET-based probes consisting of CFP-YFP as a FRET-pair, probes containing GFP and RFP variants have been developed. Such red-shifted probes allow the performance of simultaneous measurements on the single cell level with, for example, Fura-2 loaded cells and the probe expressed in the ER (Figure 5a) or mitochondria (Figure 5b). Such analysis of Ca^2+^ within the cytosol and organelles of single living cells over time allows the correlation of subcellular Ca^2+^ dynamics (Figure 5c,d) [32,48].

Considering the construction of these GEPs sensitive for Ca^2+^ based on the human variants of CaM and M13, it is difficult to claim whether these probes may interfere with, or promote, intracellular signaling cascades. Therefore, Ca^2+^ sensitive, FRET-based GEPs using Ca^2+^ binding domains of non-mammalian origin were developed. Famous examples of such Ca^2+^ sensitive probes are based on chicken skeletal muscle troponin C (TnC) located within the FP FRET-pair, bearing the advantage of using Ca^2+^ sensitive domains of non-mammalian origin [122,123,124]. TnC is a Ca^2+^ binding protein found in the thin filament of skeleton muscle cells. Upon Ca^2+^ binding to TnC, muscle contraction is initiated due to the change of the proteins conformation which is Ca^2+^ dependent [125]. Additionally, Ca^2+^ sensitive probes not directly binding Ca^2+^, but measuring the Ca^2+^ bound form of calreticulin, the major Ca^2+^ binding protein in the lumen of the ER, have been developed [126]. These indicators feature the advantage of being not directly Ca^2+^ buffering, but measuring an endogenous, Ca^2+^ bound protein. Using these probes, however, showed similar Ca^2+^ kinetics within the ER upon cell stimulation in comparison to other frequently used ER targeted, Ca^2+^ sensitive probes [32,126].

Besides the availability of a large variety of FRET-based indicators for Ca^2+^, Ca^2+^ indicators consisting of only one FP variant have been developed [41,44,127]. These single FP-based indicators can function either in an intensiometric manner where only one fluorescence signal is recorded, or allow a ratiometric read-out, indicating that, dependent on the [Ca^2+^], either two excitation maxima at constant emission, or two emission maxima at constant excitation allow the absolute quantification of [Ca^2+^]. These probes represent valuable tools for multi-channel and multi-parameter imaging due to their narrow fluorescence properties [30,41,44]. A detailed review focusing, amongst others, on single FP-based genetically encoded Ca^2+^ indicators is available [128]. A selection of FRET-based genetically encoded Ca^2+^ indicators suitable for intracellular Ca^2+^ imaging is given in Table 3.

## 4. Genetically Encoded Fluorescent Probes for Transition Metal Ions

### 4.1. Genetically Encoded Cu^+^/Cu^2+^ Indicators, Highly Sensitive Tools for Low Concentrated Ions

Copper ions (Cu^+^/Cu^2+^) represent an essential element for life due to their fundamental role as cofactors in protein functions [12]. The reversibility of the oxidation and reduction of Cu^+^ and Cu^2+^ make the function of this metal ion versatile. The cellular Cu^+^/Cu^2+^ homeostasis has to be tightly regulated, as an excess of this ion is associated with severe cellular dysfunctions and pathologies [130]. Especially diseases such as Wilson’s and Menkes disease are linked to dysregulations of the Cu^+^/Cu^2+^ uptake and are prominent examples of Cu^+^/Cu^2+^ associated pathologies [131,132]. Free Cu^+^/Cu^2+^ catalyze Fenton-type reactions, thereby producing ROS [133]. Thus it is not surprising that elevated levels of Cu^+^/Cu^2+^ are highly toxic and dysregulations of the Cu^+^/Cu^2+^ homeostasis lead to cardiovascular disorders and neurodegeneration [134,135,136]. 

In principle, Cu^+^/Cu^2+^ has high affinity to most ligands, thus the kinetically labile intracellular Cu^+^/Cu^2+^ needs to be buffered to low free concentrations to prevent its binding to other metal ion binding sites [137,138]. Therefore, many intracellular ligands for Cu^+^/Cu^2+^ exist, preventing unspecific binding [138]. As a result, only high-affinity Cu^+^/Cu^2+^ binding sites are able to compete for cellular Cu^+^/Cu^2+^, generating the Cu^+^/Cu^2+^ specificity of such proteins [36,139]. Due to the intracellular Cu^+^/Cu^2+^ buffering, it is more meaningful to consider the global Cu^+^/Cu^2+^ availability and the effective intracellular concentration, rather than the free ion concentration [36]. Several small chemical fluorescent indicators applicable for the detection of Cu^+^ have been developed recently, while only a few Cu^+^ sensitive GEPs are available [140]. 

One approach for the detection of Cu^+^ within single living cells was published in 2010 by Wegner et al., establishing a FRET-based GEP suitable for intracellular Cu^+^ measurements [36]. The probe referred to as Amt1-FRET enables Cu^+^ imaging on the single cell level. As the name suggests, the sensor is based on Amt1, a protein from *Candida glabrata (C. glabrata)*, flanked by a CFP and a YFP variant. Only amino acids 36-110 from the whole length protein were used for the design of the probe, as these amino acids represent the actual Cu^+^ binding structure. Upon Cu^+^ binding to Amt1, the probe undergoes a conformational rearrangement, yielding increased FRET and decreased donor fluorescence [36]. In *C. glabrata*, Amt1 represents a Cu^+^ binding transcription factor, inducing the expression of genes involved in the detoxification of elevated intracellular Cu^+^ levels [141]. Interestingly, the probe was not only sensitive for Cu^+^, but had a similar affinity for silver ions (Ag^+^) and a much lower affinity for Zn^2+^. Using Amt1-FRET the authors demonstrated an increase in free intracellular Cu^+^ upon incubation of cells in copper sulfate containing media, which was reversible upon addition of a Cu^+^ ligand [36]. In addition, Wegner et al. published two FRET-based GEPs in 2011 and demonstrated their suitability for Cu^+^ imaging in yeast [142]. The probes referred to as Ace1-FRET and Mac1-FRET are designed in analogy to Amt1-FRET and are based on Cu^+^ dependent transcription factors, also derived from yeast. Ace1 represents a transcription factor activating the expression of Cu^+^ detoxification and storage genes, while Mac1 senses the lower Cu^+^ level, regulating the expression of genes responsible for Cu^+^ uptake [142,143,144,145]. 

Besides Amt1-FRET, AceI-FRET and MacI-FRET, which are suitable for Cu^+^ measurements in the atto- to zeptomolar range, Yan et al. established a novel FRET-based Cu^+^ sensor in 2012 [146]. The GEP referred to as PMtb-FRET is based on the conformational change of CDC 1551 derived from *Mycobacterium tuberculosis* (*Mtb*) [147]. *Mtb* is the bacterium responsible for causing tuberculosis, a well-known disease of the lungs, but can also attack other parts of the body, including the brain, spine, kidney and arthrosis [148,149,150,151]. It is believed that especially Cu^+^-binding domains of *Mtb* cause its ability to adapt to a huge variety of challenging environments [152]. For their design of a FRET-based probe Yan et al. sub-cloned the amino acid residues 1-162, which represent the actual Cu^+^-binding domain of CDC 1551, into a CFP-YFP FRET-pair. Upon Cu^+^-binding to the peptide, FRET-efficiency is increasing, enabling a ratiometric read-out of the sensor. The probe showed higher sensitivity for Cu^+^ in the zeptomolar range. Similar to Amt1-FRET the sensor responds to other ions including Zn^2+^, Fe^2+^ and cadmium ions (Cd^2+^) [146]. Nonetheless, PMtb-FRET was not tested upon expression within single living cells in their study, hence it remains elusive whether PMtb-FRET represents a tool suitable for the visualization of intracellular Cu^+^ dynamics. 

In general, FRET-based Cu^+^-sensitive probes have only rarely been applied for intracellular Cu^+^ measurements, yet their properties and features seem highly promising for the visualization of intra- and subcellular Cu^+^ dynamics under diverse experimental setups [36,142,146,153]. A selection of FRET-based genetically encoded Cu^+^ indicators for intracellular Cu^+^ imaging is given in Table 4.

### 4.2. Genetically Encoded Zn^2+^ Indicators, A Broad Palette of Applicable Probes

Intracellular Zn^2+^ is an important micronutrient for many fundamental cellular processes, including structural functions in DNA-binding proteins, non-redox catalysis as well as neurotransmission [154,155,156]. Interestingly, around 10% of all genes code for Zn^2+^-binding motifs, demonstrating its importance in cell physiology [157]. Zn^2+^ homeostasis needs to be tightly regulated on the (sub)cellular level, as low micromolar concentrations of free Zn^2+^ can be cytotoxic [158]. Still, especially the intracellular Zn^2+^ homeostasis is poorly understood, making it an interesting target for the design of fluorescent, Zn^2+^ sensitive GEPs [159]. Notably, FRET-based probes designed for the measurement of intracellular Zn^2+^ concentrations ([Zn^2+^)_i_] need to be highly sensitive and measure Zn^2+^ concentrations ([Zn^2+^]) in the nanomolar range.

The first successful design of a genetically encoded, FRET-based indicator suitable for the detection of Zn^2+^ was achieved by van Dongen et al. in 2006 [35]. The probe is constructed from two different proteins which should interact upon Cu^+^ binding and consists of cyan and yellow FP variants fused to Atox1 or the fourth domain of copper transporting P-type ATPase ATP7B (WD4), respectively. In principle, Atox1 represents a human copper chaperone protein which targets Cu^+^ for excretion and incorporation into extracellular Cu^+^ proteins, while ATP7B is located in the Golgi membrane and represents the actual Cu^+^ transporter [160,161]. Initially, the probe referred to as CA + WY (CFP-Atox1 + WD4-YFP) was designed to sense Cu^+^, however, increases in FRET-efficiency appeared very unstable, as DTT could disrupt the Cu^+^-Atox1-WD4 complex. Interestingly, the probe responded to very low concentrations of Zn^2+^ of 10^−10^ M, also in the presence of DTT. Thus, the authors aimed to optimize the Zn^2+^ sensing behavior of the indicator and mutated the Cu^+^-binding motifs for known Zn^2+^-binding consensus sequences [35]. Nonetheless, the probe was not applied for the detection of intracellular Zn^2+^, probably due to the low dynamic range of the sensor and its lack of specificity for other ions such as cobalt ions (Co^2+^), plumb ions (Pb^2+^) and Cd^2+^. 

In 2007, van Dongen et al. optimized their first design of the Zn^2+^ sensitive probe CA + WY and designed fusion constructs of CA + WY by introducing linkers of different lengths between the two proteins. Insertion of the linkers drastically increased the dynamic range and the sensitivity of the probes referred to as CA-L2-WY containing two, CA-L5-WY containing five and CA-L9-WY containing nine GGSGGS repeats [162]. 

Dittmer et al. introduced the first FRET-based indicators for the intracellular visualization of Zn^2+^ in 2009 [163]. The authors generated a FRET-based GEP based on a CFP and a YFP variant, fused to the terminal ends of a canonical Cys_2_His_2_ zinc finger, derived from the mammalian transcription factor Zif268, a Zn^2+^ specific binding protein [163]. Structural data of Zif268 suggested an unstructured conformation in the absence of the metal ion which folds into a compact structure upon Zn^2+^ binding [164]. In mammals, zinc fingers of the Cys_2_His_2_ class recognize a wide variety of different DNA sequences and mostly represent *trans* regulators of gene expression [165]. Thus, they are fundamentally involved in development, differentiation and oncosuppression. Generally, zinc fingers represent short protein motifs consisting of two or three β-layers and one α-helix. These motifs bind Zn^2+^ which stabilizes their structure, enabling their function as transcription factors [165]. Besides their interaction with DNA, zinc fingers can also provide protein-protein and RNA-protein interactions [166,167]. However, the usage of the Cys_2_His_2_ domain derived from Zif268 within a FP FRET-pair proved suitable for the design of functional Zn^2+^ sensitive indicators. Moreover, the authors introduced mutations in the Cys_2_His_2_ domain and yielded FRET-based constructs containing His_4_, showing lower Zn^2+^ affinity but a higher dynamic range, or created a construct containing Ala_4_, lacking Zn^2+^ sensitivity. In their study, Dittmar et al. applied their novel probes for the quantification of [Zn^2+^] within the cytosol and mitochondria of single living cells with and determined [Zn^2+^] of 180 nM and 680 nM for both localizations, respectively [163].

Nowadays, a huge variety of applicable detection methods for intracellular Zn^2+^ is available [140]. One interesting approach for the development of Zn^2+^ sensitive probes represent the sensors referred to as ZinCh [168]. These sensors consist of a CFP and a YFP variant, fused via a flexible linker. The FPs within the probe possess point mutations, directly generating a Zn^2+^ sensitive pocket, yielding high FRET-efficiency in the presence of Zn^2+^, while in the absence of Zn^2+^ FRET-fluorescence is low and CFP fluorescence is increased [168]. Recently, these probes were optimized yielding eZinCh-2, suitable for the determination of intracellular [Zn^2+^] [169]. Besides the expression in the cytosol of cells, the indicator was targeted to the ER, mitochondria and insulin secreting vesicles, enabling the dynamic visualization of Zn^2+^ alterations with high spatial and temporal resolution [169]. A selection of FRET-based genetically encoded Zn^2+^ indicators suitable for intracellular Zn^2+^ imaging is given in Table 5.

## 5. Approaches for the Design of Novel Genetically Encoded Na^+^, Fe^2+^/Fe^3+^ and Mn^2+^ Indicators

Na^+^ represents the most abundant extracellular cation, involved in the maintenance of the cell membrane potential [172,173]. Na^+^ is especially important for the induction of action potentials, as it represents the first ion flowing across the membrane upon cell membrane depolarization [174]. In addition, extracellular Na^+^ concentrations ([Na^+^]_ex_) are a master regulator of the osmotic equilibrium in organisms, regulate the blood volume as well as the blood pressure [175]. Furthermore, Na^+^ functions as an important cofactor of clotting proteases such as thrombin [176]. 

Our understanding of the actual subcellular Na^+^ homeostasis and its role in regulating cell physiology and cellular functions remains, however, quite poor. Although small chemical fluorescent indicators sensitive for Na^+^ are available, these probes do not represent applicable detection methods. These fluorescent chemical Na^+^ indicators bind Na^+^ via a crown ether structure within a certain cavity [177,178,179,180,181]. However, most available probes suffer from poor specificity for Na^+^ over K^+^, and their affinity for Na^+^ appears to be dependent on [K^+^] or is far from the physiologic intracellular Na^+^ concentration ([Na^+^]) [177,181,182]. On the other hand, several probes only allow an intensiometric read-out, making quantifications of absolute [Na^+^] challenging [178,179,180,181,182]. Additionally, all of these chemical probes are difficult to target to subcellular structures such as the endoplasmic reticulum (ER), the nucleus or the Golgi apparatus, although mitochondria localized Na^+^ indicators are available [182]. Altogether, these clear disadvantages and shortcomings of chemical Na^+^ indicators demonstrate the urgent need for an applicable detection method of subcellular Na^+^. One approach to yield such optimized probes might be the development of genetically encoded, FRET-based Na^+^ indicators. Although several proteins specifically binding Na^+^ have been described, a development of GEPs for Na^+^ has mainly been hampered by the lack of Na^+^ specific binding domains, undergoing a conformational rearrangement upon Na^+^ binding [183,184,185]. The identification of such a peptide may allow the design of the first GEP sensitive for Na^+^. In addition, such GEPs will appear highly advantageous over available chemical indicators, as they can be targeted to subcellular structures and organelles, are non-toxic upon cellular expression and can also be applied in living animals [28,47]. Still the probe needs to be specific for Na^+^ over all other ions, and the Na^+^ binding should not appear in a [K^+^] dependent manner. 

Iron is fundamentally involved in important cellular functions and is required by all known living organisms [186]. In biology, iron almost exclusively exists in the form of Fe^2+^ (ferrous state) or Fe^3+^ (ferric state), although Fe^3+^ is only poorly soluble at neutral pH in aqueous media [140]. Hence, the cytosolic reduction potential favors Fe^2+^ over Fe^3+^, which is involved in ensuring the proper function of Fe^2+^-requiring enzymes [187]. These enzymes usually contain heme prosthetic groups, important for the catalysis of oxidation reactions [188]. Fe^2+^ participates in the transport of soluble gases, as it is a central component of hemoglobin and represents an essential constituent of cytochromes and other components of respiratory enzyme systems [189,190,191]. Additionally, inorganic iron is involved in redox reactions, and hence is a fundamental cofactor for enzymes such as the nitrogenase, involved in the synthesis of ammonia from nitrogen and hydrogen, hydrogenases, catalyzing the reversible oxidation of hydrogen, ribonucleotide reductase, reducing ribose to deoxyribose for DNA biosynthesis, or purple acid phosphatase, hydrolyzing phosphate esters [192,193,194,195]. Typically, cells store intracellular iron via ferritin, a Fe^2+^ binding protein [196]. 

Besides the importance of Fe^2+^ for essential biological functions, free Fe^2+^ participates in Fenton chemistry, promoting the production of reactive oxygen species (ROS) causing cell toxicity [197]. For this reason, the intracellular Fe^2+^ homeostasis needs to be tightly regulated, in order to balance the absolute cellular requirement on Fe^2+^ and toxicity induced by excess Fe^2+^ [198]. According to the importance of Fe^2+^ for fundamental cellular functions, numerous fluorescent chemical probes suitable for the detection of intracellular Fe^2+^ and Fe^3+^ have been developed. In principle, Fe^2+^/Fe^3+^ binding to these fluorescent chemical sensors changes their conformation, causing alterations of their fluorescence properties. These indicators either work in a turn-on or turn-off manner, indicating that their fluorescence is increasing or quenched upon Fe^2+^/Fe^3+^ binding, however, also ratiometric sensors have been developed [140,199,200,201,202,203].

Mn^2+^ represents an essential micronutrient for all kingdoms of life [204]. Besides its importance as a cofactor of many critical enzymes, the mitochondrial redox balance, the cellular adaption to oxidative stress, intracellular vesicle shuttling and neuron function are dependent on Mn^2+^ [205,206,207,208,209]. As with most metal ions, excess of Mn^2+^ can be toxic for cells [210]. Especially neurotoxicity is linked to Mn^2+^ accumulations and symptoms similar to those from Parkinson’s disease have been linked to disturbances of the neuronal Mn^2+^ homeostasis [209]. Considering these manifold roles of Mn^2+^ in health and disease, scientists developed chemical fluorescent indicators, suitable for Mn^2+^ imaging on the level of single cells [211]. Upon Mn^2+^ binding to these probes, their fluorescence is drastically increasing, allowing an intensiometric read-out. 

However, the development of Fe^2+^, Fe^3+^ and Mn^2+^ sensitive GEPs is still hampered by the identification of suitable ion binding domains. Interestingly, several Mn^2+^ binding proteins, especially from bacterial origin, have been described and analyzed [212,213]. Also enzymes requiring and binding Mn^2+^ in mammals have been identified [214,215]. Similarly, several iron binding sites are known, presenting opportunities for the development of Fe^2+^, Fe^3+^ and Mn^2+^ sensitive probes. However, to our knowledge, these proteins have probably either not been tested within GEPs yet or did not prove suitable for the usage in a respective protein-based fluorescent metal ion indicator.

Ideally, ion sensitive domains used within these GEPs would be highly selective for one ion over all other ions, bind the ion reversibly, repetitively and with fast kinetics. Additionally, the design of FRET-based indicators requires a protein undergoing a change in conformation upon ion binding [37]. However, other approaches such as the usage of ion dependent protein-protein interactions have also been applied and yielded functional probes [35]. One possibility to develop such GEPs applicable for the visualization of Fe^2+^, Fe^3+^ or Mn^2+^ may aim to take advantage of an unspecific ion binding by already existing probes. One could use existing probes which are not specifically binding the desired ion with low affinity and try to modify the binding site by the insertion of mutations, in order to change the ion selectivity. That such an approach may be suitable for the generation of genetically encoded Fe^2+^/Fe^3+^ and Mn^2+^ probes has been recently proven by Koay et al. who modified the specificity of a GEP by the introduction of mutations in an approved Cu^+^ sensor, thereby generating probes sensitive for Zn^2+^ [153]. 

## 6. Concluding Remarks and Outlook

This review provides an overview of genetically encoded, FRET-based biosensors, allowing the investigation of intracellular ion dynamics and concentrations in living cells with high spatiotemporal resolution. We present selected, commonly used sensors for the most important ions in cell physiology including K^+^, Mg^2+^, Ca^2+^, Cu^+^/Cu^2+^ and Zn^2+^, and address possible approaches which might yield novel GEPs suitable for the visualization of Na^+^, Mn^2+^ and Fe^2+^/Fe^3+^. The advantages of using GEPs instead of chemical fluorescent indicators are highlighted. Hence, we demonstrate remarkable examples of simultaneous applications of either two different GEPs or GEPs in combination with chemical fluorescent indicators. Both approaches drastically increase the spatiotemporal information of (sub)cellular ion signals. Summing up, we want to offer scientists an overview of GEPs suitable for the visualization of intracellular metal ions and their applications. 

However, the visualization of intracellular ion signals using FRET-based probes requires a sophisticated experimental setup, as upon excitation of the donor FP two emission wavelengths have to be recorded simultaneously [42,46,48]. Despite the refined technical requirements to measure ion signals using FRET-based sensors, their application in vivo has been demonstrated. FRET-based K^+^ sensors proved applicable for extracellular K^+^ measurements, or FRET-based Ca^2+^ indicators expressed in neurons were applied for the visualization of intracellular Ca^2+^ signals [30,216].

The broad spectral properties of the indicators often hamper their combination with other probes for multi-parameter imaging. Multiple applications of FRET-based sensors in combination with single FP-based indicators have been demonstrated [30,32,48]. As FRET-based sensors consist of two FPs, they represent relatively bulky constructs. Frequently, their targeting to subcellular compartments appears difficult e.g., into the matrix of mitochondria. However, it has been demonstrated that the fusion of multiple targeting peptide repeats can facilitate and significantly improve the targeting of these probes, resulting in their distinct subcellular localization [119].

FRET-based indicators represent the gold standard for intracellular ion concentration measurements. Their feature of a ratiometric read-out enables ion quantifications almost independent of the concentration of the probes. Nonetheless, different maturation times as well as partial sensor degradation of donor- and acceptor FPs used within FRET-based probes may have an effect on the quantification of ions using these indicators [39]. Additionally, these probes cannot be applied under hypoxic conditions, as molecular oxygen (O_2_) is required for the maturation of the FP chromophore [217].

## Figures and Tables

**Figure 1 cells-08-00492-f001:**
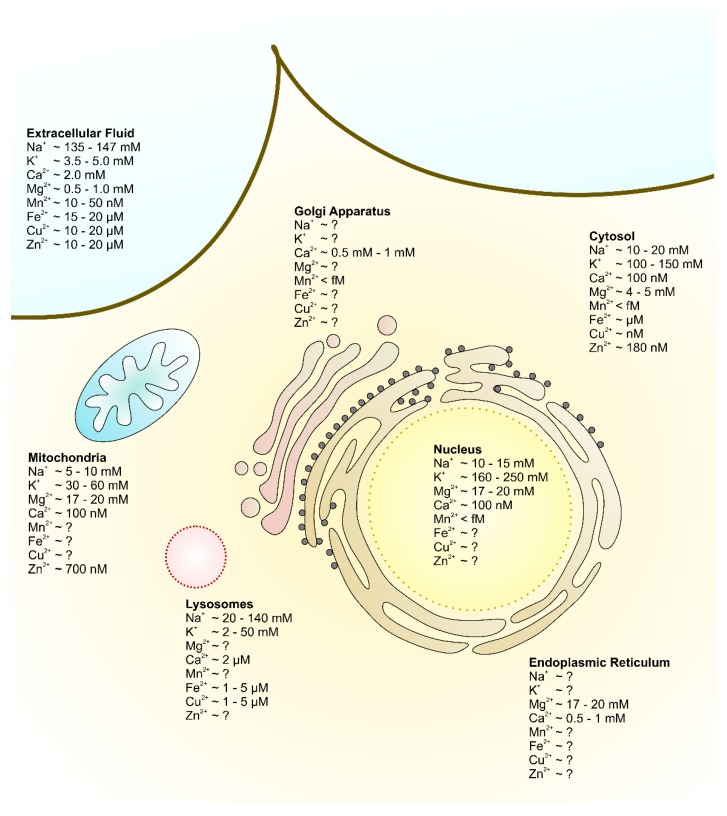
Schematic representation of extra- and intracellular metal ion distributions. Concentrations of the physiologically most relevant metal ions in the extracellular fluid (upper left, light blue) and various cellular organelles including the cytosol (upper right, light brown), nucleus (light yellow), endoplasmic reticulum (brown), Golgi apparatus (red), mitochondrial matrix (blue) and lysosomes (pink) are demonstrated.

**Figure 2 cells-08-00492-f002:**
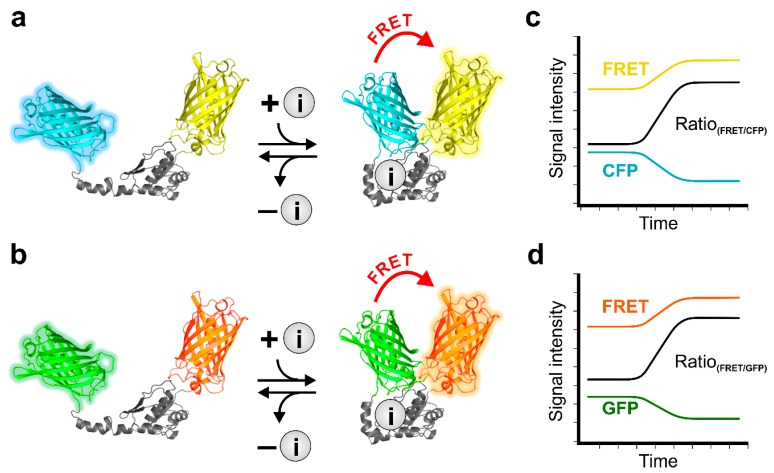
Schematic representation of the functional principle of genetically encoded probes (GEPs). Schematic representation of FRET-based indicators either containing (**a**) a CFP (cyan) and YFP (yellow) or (**b**) a GFP (green) and RFP (orange). In the ion (i, grey circles, **a** and **b**) unbound conformation FRET efficiency is low, donor fluorescence is high. Upon ion binding to the ion sensitive domain, the probes undergo a conformational rearrangement, increasing FRET-fluorescence, while the donor fluorescence, i.e., CFP or GFP, is decreasing (**a** and **b**, right panels). (**c**,**d**) schematically demonstrate the course of FRET- (yellow and orange curve, **c** and **d**), donor fluorescence (CFP, cyan and GFP, green curve, **c** and **d**), as well as the course of the resulting FRET-ratio signal of dividing FRET- by donor fluorescence (Ratio, black curves, **c** and **d**).

**Figure 3 cells-08-00492-f003:**
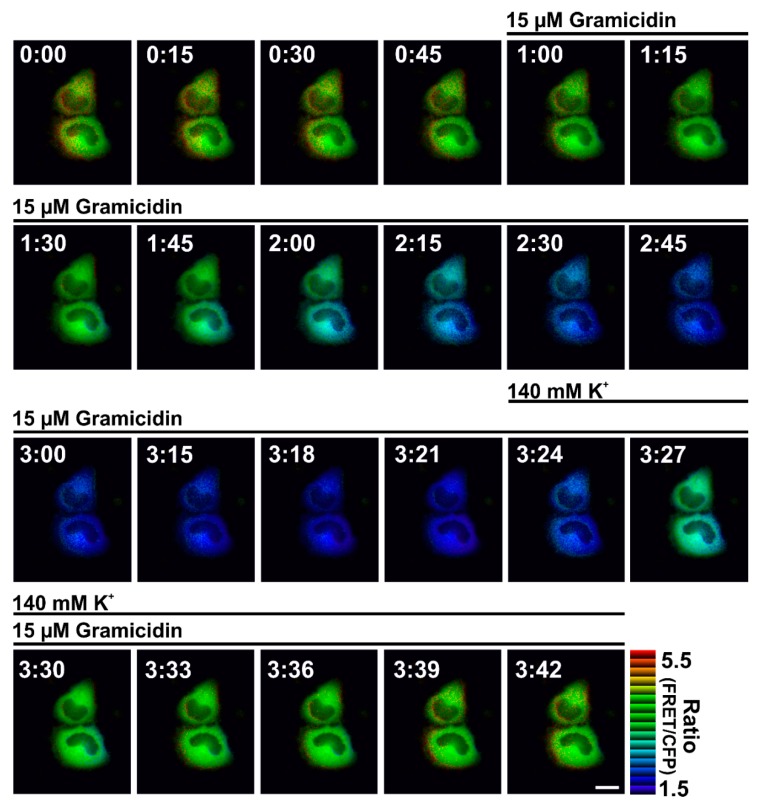
Pseudo-colored ratio images over time of HeLa cells expressing GEPII 1.0, a FRET-based K^+^ sensor. Pseudo-colored ratio images of HeLa cells expressing NES-GEPII 1.0 were treated with 15 µM gramicidin, a K^+^ selective channel forming peptide, in the absence or presence of 140 mM extracellular K^+^ as indicated. White numbers in the images represent the time in min:sec of image acquisition. Scale bar in the right lower image represents 10 µM.

**Figure 4 cells-08-00492-f004:**
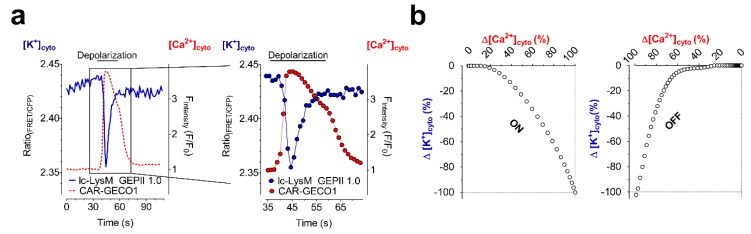
Co-expression of genetically encoded metal ion indicators allows the correlation of intracellular Ca^2+^ with K^+^ dynamics on the single cell level. (**a**) Representative single cell response of an INS-1 832/13 cell, a pancreatic β-cell, expressing lc-LysM GEPII 1.0 (blue curve, both panels), a genetically encoded, FRET-based K^+^ indicator, and CAR-GECO1, a far-red fluorescent, single FP-based Ca^2+^ sensor (red curve, both panels). At the time point indicated in the panels, cells were depolarized by the application of 70 mM K^+^. The right panel demonstrates a zoom of the area indicated in the left panel. Single time-points measured during the depolarization are indicated (right panel, blue and red circles). (**b**) Analysis of the on- (left panel) and off-kinetics (right panel) of the intracellular K^+^ and Ca^2+^ transient upon cell depolarization as demonstrated in (**a**).

**Figure 5 cells-08-00492-f005:**
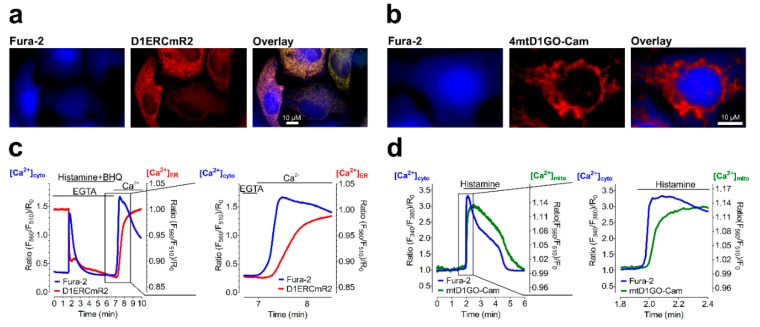
Correlation of subcellular Ca^2+^ dynamics using genetically encoded probes and Fura-2. (**a**) Fluorescence wide-field microscopy images of HeLa cells loaded with Fura-2 (blue, left image) and expressing a FRET-based, red-shifted Ca^2+^ sensitive probe targeted to the lumen of the ER (D1ERCmR2, red, middle image). Right image shows the overlay of both fluorescence images. Scale bar represents 10 µM. (**b**) Fluorescence wide-field microscopy images of a HeLa cell loaded with Fura-2 (blue, left image) and expressing a FRET-based, red-shifted Ca^2+^ sensitive probe localizing in the mitochondrial matrix (4mtD1GO-Cam, red, middle image). Right image shows the overlay of both fluorescence images. Scale bar represents 10 µM. (**c**) Representative single cell response of a HeLa cell expressing D1ERCmR2 (red curve, both panels), additionally loaded with Fura-2 (blue curve, both panels) upon cell stimulation with Histamine and BHQ, in the absence (EGTA) or presence of Ca^2+^ as indicated in the panels. Right panel shows a zoom as indicated in the left panel, demonstrating the correlation of Ca^2+^ in the cytosol and endoplasmic reticulum. (**d**) Representative single cell response of a HeLa cell expressing mtD1GO-Cam (green curve, both panels), additionally loaded with Fura-2 (blue curve, both panels) upon cell stimulation with Histamine as indicated in the panels. Right panel shows a zoom as indicated in the left panel, demonstrating the correlation of Ca^2+^ in the cytosol and the mitochondrial matrix.

**Table 1 cells-08-00492-t001:** Selection of FRET-based GEPs suitable for subcellular K^+^ imaging.

GEP	Localization	K_D_ In Vitro/In Situ	Dynamic Range	λ_Exc_ (nm)	λ_Em_ (nm)	Ref.
***GEPII 1.0***	Cytosol, Nucleus, Mitochondria, Subplasma membrane *	420 µM/820 µM	220%	430	475/525	[30]
***lc-LysM GEPII 1.0***		30.47 mM/60.95 mM	80%	430	475/525	[30]
***GEPII 2.7***	Cytosol	3.24 mM/9.09 mM	110%	430	475/525	[30]
***GEPII 2.10***		4.39 mM/10.11 mM	150%	430	475/525	[30]
***GEPII 2.15***		8.59 mM/15.59 mM	150%	430	475/525	[30]
***lc-LysM R-GEPII 1.0***		-/75.12 mM	30%	480	510/560	Unpublished
***R-GEPII 1.0***		-/3.25 mM	20%	477	510/560	Unpublished
***KIRIN1***		1.66 mM/-	130%	430	475/525	[66]
***KIRIN1-GR***		2.56 mM/-	20%	480	510/560	[66]

* indicates that the probes were fused to specific targeting sequences to obtain the given localizations, respectively.

**Table 2 cells-08-00492-t002:** Selection of FRET-based GEPs suitable for subcellular Mg^2+^ imaging.

GEP	Localization	K_D_	Dynamic Range	λ_Exc_ (nm)	λ_Em_ (nm)	Ref.
***MagFRET-1***	Cytosol	150 µM	49%	430	475/525	[34]
***MagFRET-2***		350 µM	33%	430	475/525	[34]
***MagFRET-3***		9.2 mM	58%	430	475/525	[34]
***MagFRET-4***		8.5 mM	62%	430	475/525	[34]
***MagFRET-5***		7.4 mM	74%	430	475/525	[34]
***MagFRET-6***		15 mM	50%	430	475/525	[34]
***MagFRET-7***		780 µM	38%	430	475/525	[34]
***MagFRET-8***		890 µM	56%	430	475/525	[34]
***MARIO***		7.2 mM	153%	430	475/525	[88]
***MagFRET-1 NLS***	Nucleus	150 µM	49%	430	475/525	[34]
***NLS-MARIO***		7.2 mM	153%	430	475/525	[88]

**Table 3 cells-08-00492-t003:** Selection of FRET-based GEPs suitable for subcellular Ca^2+^ imaging.

GEP	Localization	K_D_	Dynamic Range	λ_Exc_ (nm)	λ_Em_ (nm)	Ref.
***YC2.1***	Cytosol	100 nM and 4.3 µM	2	430	475/525	[27,129]
***YC3.1***		1.5 µM	2	430	475/525	[27,129]
***YC3.3***		1.5 µM	~ 1.1	430	475/525	[27,115]
***YC6.1***		110 nM	2	430	475/525	[116]
***YC3.60***		250 nM	560%	430	475/525	[117]
***D1***		800 nM and 60 µM	-	430	475/525	[118]
***D2cpV***		30 nM and 3 µM	5.3	430	475/525	[119]
***D3cpV***		600 nM	5.1	430	475/525	[119]
***D4cpV***		64 µM	3.8	430	475/525	[119]
***TN-humcTnC***		470 nM	100%	430	475/525	[123]
***TN-L15***		1.2 µM	100%	430	475/525	[123]
***TN-XL***		2.5 µM	400%	430	475/525	[124]
***LynD3cpV***	Subplasma-membrane	600 nM	5.1	430	475/525	[119]
***Cav2.2-TN-XL***		2.5 µM	-	430	475/525	[122]
***TN-L15D107ARas***		29 µM	100%	430	475/525	[123]
***H2BD1cpV***	Nucleus	800 nM and 60 µM	-	430	475/525	[120]
***4mtD3cpV***	Mitochondria	600 nM	5.1	430	475/525	[119]
***4mtD1GO-Cam***	Mitochondria	1.53 µM	-	477	510/560	[48]
***N33D1cpV***	Outer mitochondrial-membrane	800 nM and 60 µM	-	430	475/525	[120]
***Split YC7.3er***	Endoplasmic Reticulum	130 µM	-	430	475/525	[121]
***D1ER***		220 µM	-	430	475/525	[118]
***D1ERCmR2***		200 µM	-	480	510/560	[32]
***apoK1-er***		124 µM	-	430	475/525	[126]
***YC4.3ER***		800 nM and 700 µM	-	430	475/525	[118]

**Table 4 cells-08-00492-t004:** Selection of FRET-based GEPs suitable for subcellular Cu^+^ imaging.

GEP	Localization	K_D_	Dynamic Range	λ_Exc_ (nm)	λ_Em_ (nm)	Ref.
***Amt1-FRET***	Cytosol	2.5 aM	-	430	475/525	[36]
***Ace1-FRET***		4.7 aM	-	430	475/525	[142]
***Mac1-FRET***		97 zM	-	430	475/525	[142]
***eCALWY-C2M/C3M***		-	-	430	475/525	[153]
***PMtb-FRET***	-	3.31 zM	-	430	475/525	[146]

**Table 5 cells-08-00492-t005:** Selection of FRET-based GEPs suitable for subcellular Zn^2+^ imaging.

GEP	Localization	K_D_	Dynamic Range	λ_Exc_ (nm)	λ_Em_ (nm)	Ref.
***CA + WY***	Cytosol	350 pM	-	430	475/525	[35]
***Cys_2_His_2_***	Cytosol, Mitochondria *	1.7 µM	2.2	430	475/525	[163]
***His_4_***	Cytosol, Mitochondria, Plasma membrane *	160 µM	4	430	475/525	[163]
***eCALWY-1***	Cytosol, Insulin-storing granules *	2 pM	2	430	475/525	[159]
***eCALWY-2***	Cytosol	9 pM	2	430	475/525	[159]
***eCALWY-3***	Cytosol	45 pM	1.7	430	475/525	[159]
***eCALWY-4***	Cytosol	630 pM	2	430	475/525	[159]
***eCALWY-5***	Cytosol	1.8 nM	1.8	430	475/525	[159]
***eCALWY-6***	Cytosol, Insulin-storing granules *	2.9 nM	1.8	430	475/525	[159]
***ZinCh-6***	-	260 nM	~ 3.5	430	475/525	[168]
***ZinCh-9***	-	500 nM and 88 µM	4	430	475/525	[168]
***ZapCY1***	ER, Golgi Apparatus *	2.5 pM	4.15	430	475/525	[170]
***ZapSM2***	Cytosol, Nucleus *	*-*	1.1	400	510/560	[171]
***ZapSR2***	Cytosol, Nucleus *	*-*	1.2	400	510/580	[171]
***ZapOC2***	Cytosol, Nucleus *	*-*	1.1	550	565/610	[171]
***ZapOK2***	Cytosol, Nucleus *	*-*	1.1	550	565/635	[171]
***ZapCmR1***	Cytosol, Nucleus *	-	1.15	480	510/560	[171]
***ZapCmR1.1***	Cytosol, Nucleus *	*-*	1.5	480	510/560	[171]
***ZapCmR2***	Cytosol, Nucleus *	-	1.4	480	510/560	[171]
***eZinCh-2***	Cytosol, Mitochondria, ER, Vesicles *	1 nM	300%	430	475/525	[169]

* indicates that the probes were fused to specific targeting sequences to obtain the given localizations, respectively.

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
