# Peer review of "Live-Cell Imaging of Physiologically Relevant Metal Ions Using Genetically Encoded FRET-Based Probes"

_cells, 2019, doi:10.3390/cells8050492_

Round 1
Reviewer 1 Report
I found very interesting the manuscript entitled "High-resolution live-cell imaging of physiologically relevant metal ions using genetically encoded FRET-based probes". However, based on the title I expected to found more pictures from microscopy to illustrate the chosen examples. As it comes to the organization and content of the main text of manuscript, I have some comments and suggestions:
1.Some of presented proteins utilize BRET not FRET for signal readout. Authors should explain the diffrence between BRET and FRET and correct the information in the text
2.Not all Tables contain the column "localization"
3.I am not sure if all KD values are properly cited. For example, YC2.1 Kd should be 100nM, whereas for TN-L15 is 1.2 μM and for mutant TN-L15D107ARos is 29 μM .
4. I do not understand the idea to include the chapters 2.1 as well as 4.1 as contain mainly information about fluorescent chemical indicators for subcellular Na+ and Fe2+/Fe3+ and Mn2+, respectively.Thus,I think it would be better if the consideration about possibility of designing GFP sensitive for these ions were included in conclusions.
5. I recommend Authors the review by Greenwald and Mehta and Zhang (Chemical Reviews 2018 118 (24), 11707-11794)
Author Response
Dear Reviewer 1,
Please find attached our point-to-point response letter addressing your comments and inputs.
We kindly thank you for your evaluation of our manuscript and remain
Yours sincerely
Helmut Bischof

Reviewer 2 Report
Bischof et al. present a comprehensive overview over the field of genetically encoded biosensors for metal ions applicable in live cell imaging. Doing so, they cover the most relevant cellular ion species. The focus of their work is on the constructs used for ion detection. The technical side of imaging and FRET analysis is not touched. I personally would have liked a bit more on technical details; however, since this is probably beyond the scope of the manuscript, it cannot be criticized.
Here and there, a more thorough, more critical assessment of the used constructs could further strengthen the manuscript.
There are a few language flaws that should be corrected:
“In principal” should be corrected to “in principle” (several occurrences)
Line 105: “gold standard”
Line 285: “of Ca2+ as an ubiquitous...”
Line 304 “consisting of”
Line 401: “such an approach”
Author Response
Dear Reviewer 2,
Please find attached our point-to-point response letter addressing your comments and inputs.
We kindly thank you for your evaluation of our manuscript and remain
Yours sincerely
Helmut Bischof
